# Endoscopic Applications of Near-Infrared Photoimmunotherapy (NIR-PIT) in Cancers of the Digestive and Respiratory Tracts

**DOI:** 10.3390/biomedicines10040846

**Published:** 2022-04-04

**Authors:** Hideyuki Furumoto, Takuya Kato, Hiroaki Wakiyama, Aki Furusawa, Peter L. Choyke, Hisataka Kobayashi

**Affiliations:** Molecular Imaging Branch, Center for Cancer Research, National Cancer Institute, National Institutes of Health, Bethesda, MD 20892, USA; hideyuki.furumoto@nih.gov (H.F.); takuya.kato@nih.gov (T.K.); hiroaki.wakiyama@nih.gov (H.W.); aki.furusawa@nih.gov (A.F.); pchoyke@mail.nih.gov (P.L.C.)

**Keywords:** near-infrared photoimmunotherapy (NIR-PIT), cancer, endoscopy, target molecule

## Abstract

Near-infrared photoimmunotherapy (NIR-PIT) is a newly developed and promising therapy that specifically destroys target cells by irradiating antibody-photo-absorber conjugates (APCs) with NIR light. APCs bind to target molecules on the cell surface, and when exposed to NIR light, cause disruption of the cell membrane due to the ligand release reaction and dye aggregation. This leads to rapid cell swelling, blebbing, and rupture, which leads to immunogenic cell death (ICD). ICD activates host antitumor immunity, which assists in killing still viable cancer cells in the treated lesion but is also capable of producing responses in untreated lesions. In September 2020, an APC and laser system were conditionally approved for clinical use in unresectable advanced head and neck cancer in Japan, and are now routine in appropriate patients. However, most tumors have been relatively accessible in the oral cavity or neck. Endoscopes offer the opportunity to deliver light deeper within hollow organs of the body. In recent years, the application of endoscopic therapy as an alternative to surgery for the treatment of cancer has expanded, providing significant benefits to inoperable patients. In this review, we will discuss the potential applications of endoscopic NIR-PIT, especially in thoracic and gastrointestinal cancers.

## 1. Introduction

Near-infrared photoimmunotherapy (NIR-PIT) is a novel cancer therapy that utilizes antibodies conjugated with IRDye700DX (IR700), a silicon phthalocyanine dye that absorbs NIR light. When the antibody-photo-absorber conjugates (APCs) are injected intravenously, they bind to selected cancer cells that express the specific antigen on the cell membrane [1,2]. Subsequent irradiation with NIR light (about 690 nm) induces necrosis and immunogenic cell death (ICD), which is a type of cell death in which the adaptive immune system responds to the onslaught of cell-associated antigens released from damaged cancer cells [3,4,5,6]. Immediately after irradiation with NIR light, the axial ligand responsible for the hydrophilicity of the IR700 molecule dissociates from the main molecule, and the APC changes from a hydrophilic compound to a highly hydrophobic compound. This alteration in the chemical nature of APCs promotes protein and lipid aggregation, causing APC-bound cells to undergo membrane damage, rupture, and necrosis [7]. This mechanism is distinctly different from conventional photodynamic therapy (PDT) [8]. The dyes associated with PDT tend to be less specific, and normal cells take them up. Moreover, the therapeutic effect of PDT is dependent on reactive oxygen species (ROS), which has the disadvantage of causing non-specific damage within adjacent normal tissues [9]. Although some necrosis occurs with PDT, the majority of the cell killing is by apoptosis, which is non-immunogenic. In contrast, cells treated with NIR-PIT release cellular contents including both tumor neoantigens and danger signals, such as calreticulin (CRT), adenosine triphosphate (ATP), high-mobility group box 1 (HMGB1), heat shock protein (Hsp) 70, and Hsp 90, which activate immature dendritic cells (DCs) and stimulate the presentation of tumor antigens to T cells [6,10]. The activated immature DCs take up cancer-specific antigens released from the ruptured cells and transition to mature DCs, which can prime naive T cells and educate them to differentiate into cancer-specific CD8^+^ T effector cells [11,12,13].

Phase I and II clinical trials of NIR-PIT in patients with recurrent head and neck cancer (HNC), in which anti-epidermal growth factor receptor (EGFR) monoclonal antibody (mAb) conjugated to IR700 (Cetuximab-IR700) was used, showed a tolerable safety profile and promising preliminary efficacy [14]. An international phase III clinical trial of NIR-PIT for patients with recurrent HNC is currently underway [15]. The US Food and Drug Administration (FDA) approved NIR-PIT for fast track recognition. Moreover, the first EGFR-targeted NIR-PIT drug (ASP-1929; Akalux^®^, Rakuten Medical Inc., San Diego, CA, USA) and a diode laser system (BioBlade^®^, Rakuten Medical, Inc.) was conditionally approved and registered for clinical use by the Pharmaceuticals and Medical Devices Agency in Japan in September 2020.

Theoretically, NIR-PIT is most suitable for the treatment of superficial tumors, since NIR light penetrates only about 1–2 cm from the tissue surface [16]. There are several potential solutions to this challenge. First, fiber optic light diffusers placed in needle catheters can be inserted in deep tumors, and this technique has been used in clinical practice [14,17,18]. Secondly, NIR light can be delivered to a tumor during surgery [19,20]. Alternatively, NIR light could be administered endoscopically (Figure 1). We have shown in mouse models that NIR-PIT can effectively treat peritoneally disseminated gastric cancer and oral cancer after a fiber-optic diffuser is placed near the tumor during endoscopy [21,22]. In general, endoscopies are easier to perform in humans than in mice due to the body size differences. Therefore, in clinical practice, the endoscope is a promising tool to deliver NIR light to tumors within or adjacent to hollow organs in a minimally invasive and safe manner. In this review, we discuss the potential clinical applications of endoscope-based NIR-PIT for cancers arising in various organs.

## 2. Current General Use of Endoscopy in Cancer Therapy in Thoracic Cancers

Bronchoscopy plays an important role in lung cancer diagnosis and has evolved over the past few decades. Currently, the diagnosis rate of peripheral lung cancer has improved by the use of virtual bronchoscopic navigation systems (VBN), electromagnetic navigation systems (ENB), robotic bronchoscopies, and radial probe endobronchial ultrasounds (REBUS) to localize tumors [24,25,26]. Accurate staging of lung cancer by endobronchial ultrasound-guided transbronchial needle aspiration (EBUS-TBNA) determines the optimal treatment approach, and its ability to obtain robust tissue samples supports the advancement of personalized medicine in lung cancer medication therapy [27,28]. Furthermore, bronchoscopies are being used for the diagnosis of lung cancer as well as for direct therapy. A total of 13% of lung cancer patients experience complications via central airway obstructions [29]. The indications for endoscopic treatments of central airway lesions include relief (palliation) of dyspnea and airway obstruction caused by advanced cancer lesions, and curative treatment of early-stage central lung cancer. Bronchoscopic local treatment is relatively easy to learn, less invasive than surgery, and has more immediate effects than radiotherapy or chemotherapy [30]. The recommended endoscopic treatment procedure for central airway stenosis due to malignant tumor invasion is a combination of laser therapy, radiofrequency therapy, argon plasma coagulation, and stents under rigid bronchoscopy [31,32]. In the past few decades, more effective bronchoscopic treatment of patients with obstructive endobronchial tumors has been explored. Light directed therapy is not only being offered as an alternative to surgical resection for early-stage central lung cancer but has also shown promising results in the treatment of locally advanced non-small cell lung cancer (NSCLC) with airway obstruction [33,34,35,36,37]. Endobronchial intratumoral chemotherapy (EITC), the direct injection of chemotherapeutic agents via a needle catheter into a bronchial tumor, has been reported to be very effective at reducing airway obstruction [38]. In addition, cryotherapy and brachytherapy can also be used, delivered bronchoscopically to relieve symptoms [39]. Recently, endoscopic therapy has become an alternative treatment option for early-stage peripheral lung cancer. Surgery is the standard treatment for early NSCLC. However, elderly patients and patients who are inoperable due to complications, such as respiratory dysfunction require less invasive treatments. Bronchoscopic treatment is the preferred treatment for these poor surgical candidates. Currently, bronchoscopic cryotherapy, transbronchial brachytherapy, bronchoscopy-guided radiofrequency ablation (RFA), and PDT are being actively investigated for early-stage peripheral lung cancer [40,41,42,43,44]. 

## 3. Application of Endoscopic NIR-PIT for Thoracic Cancers

The lung is an ideal organ for light therapy because light can easily penetrate through aerated lung tissue without significant attenuation. Bronchoscopes can easily direct NIR light to central airway lesions. Synchronous or metachronous tumors are not uncommon in central early-stage lung cancers [37]. Unlike radiation therapy, NIR-PIT can be used to treat multiple lesions in a single session and has no restrictions on repeat treatments [2]. Furthermore, in mouse models, cancer cell-targeted NIR-PIT results in an antitumor immune response both locally and at distant sites. Repeated NIR-PIT could act as immune boosters [11,12,45,46,47,48]. Therefore, the antitumor immune activity and immunological memory of NIR-PIT may be useful against multiple primary lung cancers. On the other hand, treatment of central airway stenosis caused by malignancy requires urgent procedures. Both PDT and EITC take a certain amount of time to be effective due to the slower kinetics of apoptosis-induced cell death [49,50]. NIR-PIT induces rapid (within minutes) necrosis and ICD immediately after exposure to NIR light [6,47,51,52]. In addition, many photosensitizers used for PDT are activated by light in the visible range, which penetrates only a few millimeters into the tissue. NIR light used in NIR-PIT can penetrate 1–2 cm into tissue and perhaps even deeper in the aerated lung where there is little light attenuation. This makes it attractive as a treatment to reduce the volume of a tumor protruding into the airway. Therefore, NIR-PIT will be a useful treatment option for central lung cancer in the future. Recent bronchoscopic technology is capable of delivering NIR light to almost any intrapulmonary lesion. NIR-PIT could become a convenient treatment for early-stage peripheral lung cancer in inoperable patients who need minimally invasive treatment because it is a local therapy with minimal side effects and no damage to surrounding normal tissues. Moreover, NIR light can be delivered transbronchially to the chest wall. Pleural dissemination and associated malignant pleural effusions, a fatal complication of lung cancer, occur frequently in patients with advanced disease [53]. Chemotherapy is the only viable treatment for NSCLC with pleural dissemination, but the prognosis is poor [54]. Although NIR-PIT monotherapy is unlikely to be curative, it has the potential to provide local control of the disease. Previous studies have shown that tumors treated with NIR-PIT had an immediate but short-lived increased permeability to nano-sized drugs. Therefore, NIR-PIT followed by chemotherapy could improve the micro-distribution of drugs within the tumor, leading to more uniform responses [55]. On the other hand, malignant pleural mesothelioma (MPM) usually progresses by local progression in the thoracic cavity rather than by distant metastasis [56]. Therefore, NIR-PIT offers minimally invasive local control by applying NIR light to residual postoperative or locally recurrent lesions. NIR-PIT may become an important adjuvant to surgery and chemotherapy. Thus, NIR-PIT and bronchoscopy are highly compatible with each other, and their combined use could expand the field of non-invasive thoracic cancer treatments.

## 4. Molecular Targets for NIR-PIT in Thoracic Cancers

### 4.1. Lung Cancer 

Lung cancer is composed of NSCLC (80–85%) and small cell lung cancer (SCLC, 15–20%) [57,58]. NSCLC includes various histological types, such as adenocarcinoma, squamous cell carcinoma, and large cell carcinoma. Each cancer type has unique cell surface markers that could be targeted for NIR-PIT. Below, possible targets for NIR-PIT of lung cancer are discussed (Table 1).

#### 4.1.1. EGFR

EGFR, belonging to the erythroblastosis oncogene B (erbB) family, is a transmembrane tyrosine kinase receptor [59]. EGFR is a molecule that plays an important role in cell survival, proliferation, metastasis (migration, adhesion, and invasion), and angiogenesis [60]. EGFR-targeted therapies using both small molecules and antibodies are already in clinical use for many cancer types, including NSCLC [61,62,63]. EGFR overexpression has been identified in 40–80% of NSCLCs, especially in squamous cell carcinoma where it frequently is expressed at high levels [64,65]. NIR-PIT and panitumumab (antibody targeting EGFR)-IR700 conjugated inhibited tumor growth in a transgenic mouse model of spontaneous lung cancer, which expressed human EGFR [66].

#### 4.1.2. Human Epidermal Growth Factor Receptor 2 (HER2)

Receptor tyrosine kinase erbB-2, also known as HER2, is a member of the EGFR family. Heterodimerization of this receptor with other members of the EGFR family, usually by overexpression of HER2, leads to autophosphorylation of tyrosine residues in the cytoplasmic domain of the heterodimer and initiates various signaling pathways leading to cell proliferation and tumorigenesis [67]. HER2 is highly expressed in about 2.5% of NSCLC [68]. HER2-targeted NIR-PIT showed antitumor effects and significantly improved survival in a HER2-expressing mouse model of lung metastasis [51,69]. It also led to a reduction in tumor volume and long-term survival in a model of NSCLC with pleural dissemination. Although HER2 is expressed in only a minority of lung cancers, HER2-targeted NIR-PIT has demonstrated efficacy against lung metastatic lesions and pleural disseminated lesions in preclinical experiments.

#### 4.1.3. Cancer Stem Cell Marker (CD44)

CD44, an essential transmembrane glycoprotein and extracellular matrix receptor, is also a cancer stem cell marker [70]. CD44 plays an important role in the invasion and metastasis of cancer cells and in basic biological processes, such as lymphocyte homing, hematopoiesis, inflammation, wound healing, and apoptosis [71]. CD44 is expressed in a variety of epithelial malignancies, and in NSCLC, overexpression is associated with poor prognosis [72]. CD44 positivity is present in 38% of SCLC patients and is, therefore, a promising target for NIR-PIT in NSCLC and SCLC [73].

#### 4.1.4. Carcinoembryonic Antigen (CEA)

CEA is one of the most commonly used tumor markers, first found overexpressed in colorectal cancer and subsequently in other cancers of the gastrointestinal tract, as well as other organs [74]. CEA is expressed not only in NSCLC but also in SCLC [75]. CEA levels in lung adenocarcinoma are higher than in squamous cell carcinoma, and CEA levels in SCLC are lower than in adenocarcinoma. CEA can be used to diagnose lung cancer, particularly adenocarcinoma [76]. 

#### 4.1.5. Podoplanin (PDPN)

PDPN is a cell surface mucin-like glycoprotein that plays an important role in the development of the alveoli, heart, and lymphatic vascular system [77]. PDPN is associated with malignant progressions, such as metastasis and invasion, and can be a therapeutic target. PDPN is overexpressed in 40% of lung squamous cell carcinomas [78]. PDPN can be useful as a target molecule for lung squamous cell carcinoma.

#### 4.1.6. Mesothelin (MSLN)

MSLN is a cell surface glycoprotein and tumor differentiation antigen expressed in some malignancies, such as mesothelioma [79]. MSLN can be used as a systemic diagnostic biomarker because it is released from the cell surface of biological fluids, such as serum, plasma, and pleural effusion [80]. MSLN is highly expressed in many epithelial carcinomas, including lung cancer. Lung adenocarcinoma expresses MSLN in 41% of cases while squamous cell carcinoma expresses it in 28% of cases, and it is not expressed in SCLC [81]. Recently, anti-MSLN mAbs have been developed and are being evaluated in clinical studies [82,83]. MSLN is, therefore, a potential target molecule, not only for MPM but also for NSCLC.

#### 4.1.7. Delta-like Ligand 3 (DLL3)

DLL3 is a ligand of the Notch signaling pathway. It is specifically expressed on the cell surface of SCLC and not in normal tissues. Currently, DLL3 is expected to be a therapeutic target molecule for SCLC [84]. However, the first antibody targeting DLL3, rovalpituzumab tesirine, was terminated in August 2019 because of the negative TAHOE (NCT03061812) and MERU (NCT03033511) clinical trials [85]. DLL3-targeted NIR-PIT, however, showed remarkable antitumor effects in SCLC xenograft models [86]. DLL3-targeted NIR-PIT with rovalpituzumab can be easily translated to clinical treatment as a means of repurposing this antibody [87].

#### 4.1.8. G Protein Receptor 87 (GPR87)

GPR87 is a member of the G protein-coupled receptor family and is overexpressed in many tumor tissues, including NSCLC. GPR87 overexpression has been found to promote cancer cell proliferation, survival, and tumorigenesis, and is associated with a poor prognosis [88,89]. A total of 51.2% of NSCLCs and 37.5% of SCLCs were GPR87 positive, and it was more frequent in squamous cell carcinoma than in adenocarcinoma [90]. GRP87-targeted NIR-PIT demonstrated therapeutic effects in mice bearing human lung cancer cells with pleural involvement [91]. GPR87 is, therefore, a promising target for NIR-PIT in NSCLC and SCLC.

#### 4.1.9. Programmed Death-Ligand 1 (PD-L1)

Several solid tumor types, including NSCLC, express PD-L1 to generate an immunosuppressive tumor microenvironment and avoid T cell cytolysis [92]. Some reports indicate that the expression of PD-L1 in NSCLC is 49–95% [93,94,95,96,97]. The PD-L1 positive rate in adenocarcinoma was slightly higher than in squamous cell carcinoma (65.2% and 44.4% respectively). However, PD-L1 expression in SCLC is relatively low [98]. Overexpression of PD-L1 in lung cancer may contribute to poor prognosis by permitting immune evasion by inhibiting the maturation of tumor-infiltrating DCs [99]. Patients with NSCLC overexpressing PD-L1 have a response rate of 67–100%, while patients with PD-L1-negative NSCLC have a response rate of 0–15% [92]. PD-L1-targeted NIR-PIT using avelumab (a fully humanized IgG1 anti-PD-L1 mAb) as the targeting agent, significantly inhibited tumor growth and prolonged survival in a xenograft model [100]. NIR-PIT utilizing PD-L1 as the targeting antigen might be successful in treating cancers that have not previously responded to monotherapy with the antibody alone.

#### 4.1.10. Multidrug Resistance Protein 1 (MRP1)

MRP1, one of the ATP-binding cassette (ABC) transporter proteins, causes decreased drug accumulation in cancer cells. Clinically, MRP1 levels are elevated in many cancer types, including NSCLC, and are also overexpressed in drug-resistant SCLC cells [101,102]. MRP1-targeted NIR-PIT showed excellent antitumor activity in a xenograft mouse model that used chemotherapy-resistant SCLC H69AR cells, which overexpress MRP1. ABC transporter may, therefore, be a new target for NIR-PIT therapy in multidrug-resistant lung cancer [103].

### 4.2. Malignant Pleural Mesothelioma (MPM)

MPM is a highly invasive tumor that originates from the pleural surface. Most MPM patients are considered inoperable at diagnosis and have a poor prognosis [104]. For some patients, surgery is a possible treatment option, but pleurectomy and extrapleural pneumonectomy are very invasive and burdensome, with a postoperative complication rate of 70% and a postoperative mortality rate of 20% [105]. Chemotherapy options are also limited for MPM [106,107]. Therefore, new treatment strategies for MPM are urgently needed. NIR-PIT for MPM has shown success in preclinical experiments using antibodies against several surface molecules expressed on MPM cells, offering a potentially new therapeutic option.

#### 4.2.1. PDPN

Approximately 80% of resected MPM specimens were positive for PDPN. Antibodies against PDPN (D2-40) have been used as the targeting molecule for MPM [108]. PDPN-targeted NIR-PIT using the anti-PDPN antibody, NZ-1, showed tumor suppression in both xenograft and orthotopic MPM models in immunodeficient mice [109]. 

#### 4.2.2. MSLN

MSLN is expressed on most mesotheliomas [110]. In xenograft mouse models, MSLN-targeted NIR-PIT has shown antitumor effects and prolonged survival [111]. MSLN is therefore an excellent target molecule for MPM. 

## 5. Current General Use of Endoscopy in Cancer Therapy in Gastrointestinal Cancers

Gastrointestinal endoscopy has undergone a remarkable expansion in recent years [112]. An endoscopy is routinely used for diagnosis and for treating various gastrointestinal disorders. When it was initially developed, the main purpose of the platform was to diagnose and observe mucosal abnormalities; however, endoscopic therapeutic procedures have greatly expanded. For instance, two common endoscopic procedures are cauterization of gastrointestinal bleeding sites and sclerotherapy of esophagogastric varices [113,114,115]. The first percutaneous endoscopic gastrostomy (PEG) was performed in 1979 and since then enteric access has become one of the most common applications for endoscopic treatment worldwide [115,116,117]. Beginning in the 1990s, endoscopic intestinal stenting using self-expanding metal stents enabled minimally invasive treatments for malignant esophageal strictures and malignant gastric outlet obstructions [118,119]. The application of an endoscopy now also includes the evaluation and treatment of biliary and pancreatic duct disorders using endoscopic retrograde cholangiopancreatography (ERCP) [115,120,121]. Natural orifice translumenal surgery (NOTES) enables access to the lumen of various hollow organs (gastrointestinal, vagina, or bladder) via the peritoneum using an endoscope [115,122,123]. This evolved to peroral endoscopic myotomy (POEM) as a technique to divide the esophageal muscles in cases of achalasia [124,125].

Endoscopes equipped with charge-coupled devices (CCDs) are now capable of capturing digital images of more than one million pixels, resulting in high-resolution images that allow for easy distinction of details and improved diagnostic accuracy [112,126]. Furthermore, with advances in endoscopic CCD cameras, endoscopic ultrasound, and endoscopic instrumentation, the diagnostic accuracy of endoscopy has dramatically improved compared to a conventional system. Endoscopic resection techniques, which are broadly classified into two categories, endoscopic mucosal resection (EMR) and endoscopic submucosal dissection (ESD), have greatly expanded in recent years, enabling the endoscopic resection of laterally spreading tumors [112,127,128]. Currently, an endoscopic resection (EMR/ESD) is recommended by several societies for superficial cancers in the gastrointestinal tract, such as those in the esophagus, stomach, and colorectal lesions [129,130,131]. 

## 6. Application of Endoscopic NIR-PIT for Gastrointestinal Cancers

EMR/ESD is minimally invasive and cosmetically superior to conventional therapy; however, there is a risk of serious complications, such as bleeding and gastrointestinal perforation [126,127,131,132]. Moreover, unresectable areas arise from the inability to reach all anatomic regions endoscopically or because the tumors invade the deeper muscular layers, making endoscopic resection difficult to perform safely. NIR-PIT may create an opportunity to expand the use of endoscopic therapy. Endoscopic NIR-PIT using a cylindrical light diffuser with an NIR light laser system in a murine disseminated gastric cancer model, which overexpressed HER2, demonstrated the potential of this therapy [21]. One day after APC (Trastuzumab-IR700) administration, the endoscope was inserted into the abdominal cavity through a small lower abdominal incision, and NIR laser light was directed at the peritoneal cavity. The cylindrical light diffuser was passed through the working channel of the endoscope. NIR-PIT induced a significantly greater therapeutic effect compared to the control group [21]. Thus, endoscopic NIR-PIT is not only possible, but appears effective in animal models. Furthermore, NIR light can be applied from within the intestinal lumen and can reasonably be expected to be effective to a depth of 1–2 cm [16]. This suggests that NIR-PIT can be applied to treat tumors that cannot be resected endoscopically for technical reasons, or because the tumor invades too deeply to be resected completely. Moreover, NIR light irradiation kills only APC-bound target cells, sparing nearby cells that have no (or minimal) expression of the target antigen [1,2]. For this reason, NIR-PIT spares normal tissue, induces an immune response in tumors and, therefore, has a great advantage for treating gastroenterological cancers.

## 7. Target Molecules for NIR-PIT in Gastrointestinal Cancers

A large body of pre-clinical research on gastrointestinal NIR-PIT has been performed [13,133,134,135]. In brief, this work demonstrates that many types of cancer cells can be selectively killed by targeting unique antigens overexpressed in the tumor [64]. Several candidates for NIR-PIT targeting are discussed below. 

### 7.1. Esophageal Cancer

Esophageal cancer is the 11th most common cancer and the 6th most common cause of cancer death in the world [136]. Esophageal squamous cell carcinoma (ESCC) has a high prevalence in East Asia, Africa, and southern Europe, while esophageal adenocarcinoma (EAC) has a high prevalence in North America and western Europe [137]. These geographical differences indicate that environmental exposures, ethnicity, genetic factors, and lifestyle are involved in the development of ESCC. However, the esophagus is one of the sites where NIR-PIT can be readily applied endoscopically (Table 2). 

#### 7.1.1. EGFR and HER2

EGFR or HER2-targeted NIR-PIT has induced significant treatment effects in esophageal cancer cells [138]. Several studies reported high EGFR expression in most esophageal cancer patients (>70% in ESCC, 30–60% in EAC); therefore, EGFR-targeted NIR-PIT is ideally suited to treating esophageal cancer and human trials are underway.

#### 7.1.2. Cancer-Associated Fibroblasts (CAFs)

Besides cancer cells, CAFs are the most abundant cell populations in the tumor microenvironment and play an important role in tumor progression, angiogenesis, metastasis, drug resistance, and immunosuppression [139,140,141,142]. Anti-tumor effects were observed after applying NIR-PIT targeting fibroblast activation protein (FAP), a specific surface marker of CAFs [140,141]. This FAP-targeted NIR-PIT may be valuable for cancers with significant amounts of stroma, especially some types of esophageal and pancreatic cancers [139,143]. Interestingly a new positron emission tomography (PET) agent based on FAP may be useful in identifying suitable patients for FAP-targeted NIR-PIT.

### 7.2. Gastric Cancer

Gastric cancer remains a cancer of global importance, ranking fifth in incidence and fourth in mortality worldwide, with (an estimated) more than 1 million new cases and 769,000 deaths in 2020 [136]. Although it is well-known that chronic *Helicobacter pylori* infection is a major cause of gastric cancer outside the gastric cardia, an increase in gastric cancer (both cardial and non-cardial) in younger adults (under 50 years-old) has been observed [144] (Table 2).

#### 7.2.1. EGFR and HER2

In mouse models, HER2-targeted NIR-PIT led to significant tumor regression in both xenografts and peritoneal carcinomatosis, and the combination of NIR-PIT and conventional chemotherapy showed a synergistic effect [145,146]. From the results of the ToGA trial, the proportion of HER2-positive gastric cancers was present in 22.1% of patients [147]. Immunohistochemical analyses demonstrated that EGFR was expressed in about 50% of gastric cancer patients [148,149]. Thus, EGFR/HER2-targeted NIR-PIT is a promising approach for gastric cancer patients. 

#### 7.2.2. CEA

CEA is a membrane-bound glycoprotein expressed in gastrointestinal neoplasms and other adenocarcinomas and has already been employed as a tumor marker for various cancers [74,149,150]. CEA-targeted NIR-PIT demonstrated highly specific antitumor effects for CEA-expressing gastric cancer in both in vitro and in vivo models [151]. 

#### 7.2.3. Cadherin-17 (CDH-17)

CDH-17 (also known as CA17) is a cell surface biomarker specific for gastrointestinal cancers and is considered to be more sensitive than keratin-20 (CK20) and caudal type homeobox-2 (CDX2), which is used to diagnose tumors [152]. CDH-17 plays an essential role in cancer cell proliferation, cell adhesion, migration, and invasion; therefore, CDH-17 could be an ideal target molecule for gastric cancer therapy [153]. CDH17-targeted NIR-PIT suppressed tumor progression in a xenograft model using a CDH-17-expressing gastric cancer [154].

### 7.3. Colorectal Cancer

Colorectal cancer is the third most common cancer in terms of incidence, and the second most common in terms of mortality [136]. Incidence rates have been steadily rising, especially in people younger than 50 years in many countries, reflecting lifestyles and dietary changes, such as increased intake of animal-derived foods, sweetened foods, and a shift to more sedentary lifestyles, with decreased physical activity, and increased excess body weight [155]. 

#### 7.3.1. EGFR 

Between 44 and 97% of patients with colorectal cancer overexpress EGFR on immunohistochemical analysis. Anti-EGFR antibodies, such as cetuximab and panitumumab, are effective therapies and have already been approved by the FDA as first-line therapy for colorectal cancer [148,156,157]. Therefore, NIR-PIT utilizing the anti-EGFR antibody might be successful in colorectal cancer.

#### 7.3.2. CEA

CEA is recognized as an excellent tumor marker of colorectal cancer; a high expression of CEA is associated with a poor prognosis in colorectal cancer patients [158]. As described before, CEA-targeted NIR-PIT has selective cytotoxicity for cancer cells in several animal models and, therefore, has great potential in treating colorectal cancer [20,151,159].

#### 7.3.3. Glycoprotein A33 Antigen (GPA33)

GPA33 was discovered as a novel membrane protein and is highly upregulated in more than 95% of colorectal cancers but is not expressed in adjacent normal cells [160,161]. Recently, it was reported that GPA33-targeted NIR-PIT induced cell-specific cytotoxicity against GPA33-positive colorectal cancer cells in xenograft models [162]. Thus, GPA33-targeted NIR-PIT could be a promising cancer therapy for colorectal cancer patients.

### 7.4. Hepatobiliary and Pancreatic Cancer

It is extremely challenging to treat hepatic, biliary, and pancreatic tumors in general, let alone through endoscopy. This is because these malignant tumors are located in regions usually inaccessible to today’s endoscopes. Even if the endoscope can be placed it is challenging to secure a space where treatment devices can be placed through an endoscope without dislodging the scope. However, because NIR-PIT does not require instrumentation, and only an NIR light diffuser is placed near the tumor transendoscopically, it could theoretically be applied to many types of endoscopy-accessible tumors. For instance, using a standard ERCP setup, it is possible to insert the light diffuser into bile or pancreatic ducts, and to irradiate tumors. This suggests that liver cancer, biliary tract cancer, and pancreatic cancer are also potential candidates for NIR-PIT. Several candidate molecular targets for these tumors are described below.

#### 7.4.1. EGFR

Similar to other gastrointestinal tumors, overexpression of EGFR is reported in more than 60% of pancreatic cancers and 30–60% of cholangiocarcinoma patients based on immunohistochemical analysis [59,163,164]. EGFR-targeted NIR-PIT is an excellent candidate for hepatobiliary and pancreatic cancers.

#### 7.4.2. CEA

CEA is not expressed in normal pancreatic tissue, while it is frequently expressed in pancreatic cancer. CEA-targeted NIR-PIT, using a pancreatic cancer cell line, has shown significant tumor suppression and prolonged survival in an orthotopic tumor model without side effects [20]. Furthermore, the combination of surgery and CEA-targeted NIR-PIT significantly reduced the recurrence rate after surgery in a patient-derived, orthotopic xenograft nude mouse model [159].

#### 7.4.3. CDH-17

In vitro and in vivo experiments using pancreatic cancer cell lines have demonstrated that CDH-17-targeted NIR-PIT could induce specific cytotoxicity against CDH-17-expressing cells and inhibit tumor progression in a xenograft model of pancreatic cancer [154]. This suggested that CDH17-targeted NIR-PIT was a possible treatment in pancreatic cancers.

#### 7.4.4. Glypican-3 (GPC3)

GPC3 is a 65 kDa membrane-bound heparin sulfate proteoglycan that is expressed in fetal liver, lung, and kidney [165]. GPC3 is an interesting target for hepatocellular carcinoma (HCC) patients, because it is highly expressed in HCC, while it is not expressed in normal tissues [166]. GPC3-targeted NIR-PIT demonstrated antitumor effects in a GPC3 positive xenograft model.

#### 7.4.5. Tumor-Associated Calcium Signal Transducer 2 (TROP2)

TROP2 is a 46-kD glycoprotein with multiple intracellular roles, notably in cytosolic calcium transfer, which is expressed in the malignant cells [167]. TROP2 has been observed in pancreatic cancer and cholangiocarcinoma [168]. NIR-PIT utilizing anti-TROP2 antibody demonstrated specific cell killing against TROP2-expressing cells and the suppression of tumor growth in a xenograft model of pancreatic cancer and cholangiocarcinoma [169]. Thus, TROP2 is an interesting target molecule for hepatobiliary-pancreatic NIR-PIT.

### 7.5. Gastrointestinal Stromal Tumors (GISTs)

NIR-PIT may also be applicable to non-epithelial malignant tumors. GISTs are the most common mesenchymal tumors of the gastrointestinal tract (stomach, 55.6%; small bowel, 31.8%; colorectum, 6.0%, esophagus, 0.7%) [170,171]. More than 90% of GISTs overexpress c-kit (CD117) on their cell membranes, making it a molecular target for NIR-PIT. Xenograft tumors in a GIST model regressed after c-KIT-targeted NIR-PIT using an anti-CD117 antibody-based APC [172]. Thus, endoscopy could be used to deliver light for NIR-PIT in patients with GIST in the GI tract. Admittedly many GISTs have metastasized at the time of diagnosis and NIR-PIT might not be useful in these cases.

## 8. Conclusions

Cancer treatments continue to evolve toward less invasive, safer, and more selective/effective treatments. NIR-PIT is an innovative treatment that specifically attacks antigen-bearing cancer cells, leading to minimal side effects. NIR-PIT delivered endoscopically has the potential to treat a variety of cancers in the lung and GI tract by targeting cancer cells and immune cells. In addition, endoscopic NIR-PIT may be able to deliver NIR light via hollow organs to deep solid organs, such as the lung/mediastinum, biliary tract/liver, and duodenum/pancreas.

## Figures and Tables

**Figure 1 biomedicines-10-00846-f001:**
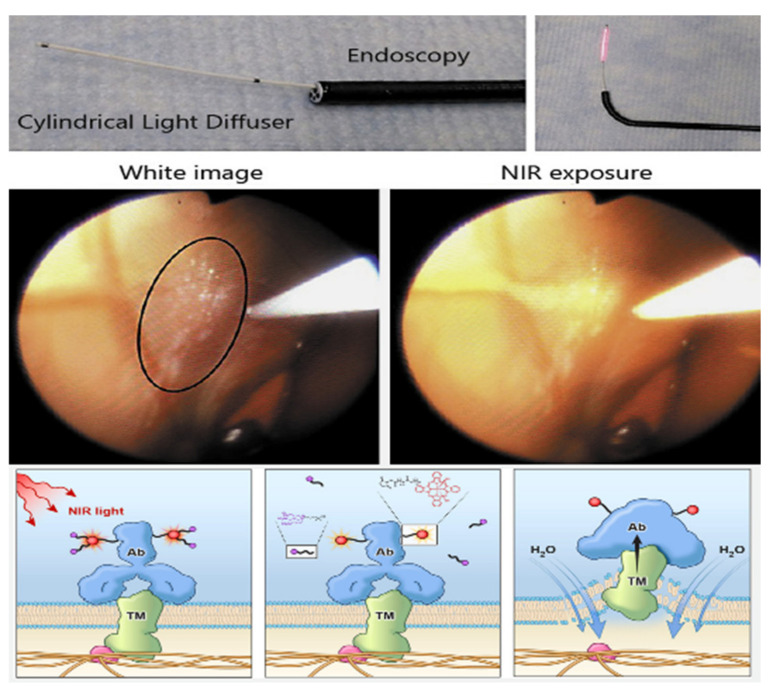
Endoscopes can deliver light deep into the hollow organs of the body. APCs bind to target molecules on the cell surface and with exposure to NIR light, cause cell death by ligand release reaction and membrane aggregation. Adapted with permission from ref. [7], copyright year 2018, ACS Cent Sci. and ref. [23], copyright year 2020, Karger.

**Table 1 biomedicines-10-00846-t001:** Target Molecules for lung cancer in NIR-PIT targeting cancer cells.

	Target Moleclue									
	EGFR	HER2	CD44	CEA	PDPN	MSLN	DLL3	GPR87	PD-L1	MRP1
NSCLC	◯	△	◯	◯	◯	◯		◯	◯	◯
	Sq > Ad			Ad	Sq			Sq > Ad	Sq > Ad	
SCLC			◯	◯			◯	◯	△	◯
MPM					◯	◯				

NSCLC, non-small cell lung cancer; SCLC, small cell lung cancer; EGFR, epidernal growth factor receptor; HER2, human epidermal growth factor receptor-2; CEA, carcinoembryonic antigen; PDPN, podoplanin; MSLN, mesothelin; DLL3, delta-like Ligand 3; GPR87, G protein-coupled receptor 87; PD-L1, programmed death-ligand 1; Multidrug Resistance Protein 1, MRP; ◯, sufficient expression; △, weak expression; Sq, squamous cell carcinoma; Ad, adenocarcinoma; MPM, malignant pleural mesothelioma.

**Table 2 biomedicines-10-00846-t002:** Target Molecules for each gastrointerstinal cancer in NIR-PIT targeting cancer cells.

	Target Molecules											
	EGFR	HER2	CD44	CEA	PDPN	MSLN	GPA33	TROP2	CDH-17	PD-L1	GPC-3	c-KIT
Gastrointestinal Ca.												
Esophageal Ca.	◯		◯	△	◯					◯		
Gastric Ca.	◯	◯	◯	◯			◯	◯	◯	◯		
Colorectal Ca.	◯	△	◯	◯			◯	◯	◯	◯		
Hepatic cell Ca.			◯							◯	◯	
cholangiocarcinoma			◯	◯				◯		◯		
Pancreatic Ca.	◯	◯	◯	◯		◯	◯	◯	◯	◯		
GIST												◯

Ca., cancer; EGFR, epidernal growth factor receptor; HER2, human epidermal growth factor receptor-2; CEA, carcinoembryonic antigen; PDPN, podoplanin; MSLN, mesothelin; GPA33, glycoprotein A33 antigen; TROP-2, tumor-associated calcium signal transducer 2; CDH-17, cadherin-17; PD-L1, programmed death-ligand 1; DLL3, delta-like protein 3; GPC-3, glypican-3; PSMA, prostate-specific membrane antigen; CLA, cutaneous lymphocyte antigen; GIST, Gastrointestinal stromal tumors; ◯, sufficient expression; △, weak expression.

## Data Availability

No new data were created or analyzed in this study. Data sharing is not applicable to this article.

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
