# Peer review of "Endoscopic Applications of Near-Infrared Photoimmunotherapy (NIR-PIT) in Cancers of the Digestive and Respiratory Tracts"

_biomedicines, 2022, doi:10.3390/biomedicines10040846_

Round 1
Reviewer 1 Report
Authors well summarized the potential of NIR-PIT in cancers of the digestive and respiratory tracts. Although there still have been some challenges in hepatobiliary and pancreatic cancer for the therapy, it is promising therapy for cancer. My minor comments are below.
I could not find any descriptions about figure 1 in main text although legend for figure 1 was found.
Figure 1 is ectopically located in the first page. This should be deleted.
Author Response
Authors well summarized the potential of NIR-PIT in cancers of the digestive and respiratory tracts. Although there still have been some challenges in hepatobiliary and pancreatic cancer for the therapy, it is promising therapy for cancer. My minor comments are below.
I could not find any descriptions about figure 1 in main text although legend for figure 1 was found.
Figure 1 is ectopically located in the first page. This should be deleted.
(Response) We have added the legend of Fig 1 in line 72. Fig 1 in the first page is the graphical abstract that requires for the article format of Biomedicines.
Reviewer 2 Report
The whole review is well written and nicely readable. Introduction is appropriate and the whole article contains enough information even for newbies in the field - so the article may be of interest to a wider audience.
I recommend just minor revisions of the formulations. Such as:
In "4. Target molecular for NIR-PIT in thoracic cancers" shouldn't it be "Molecular target for..." or "Target molecules for..."?
Same applies in: "7. Target molecular for NIR-PIT in gastrointestinal cancers"
Similarly: Table 1. "weakly expression" shouldn't it be "weak expression" or "weakly expressed"?
I have no serious complaints about the presented publication and I recommend it for publication in Biomedicines.
Author Response
The whole review is well written and nicely readable. Introduction is appropriate and the whole article contains enough information even for newbies in the field - so the article may be of interest to a wider audience.
I recommend just minor revisions of the formulations. Such as:
In "4. Target molecular for NIR-PIT in thoracic cancers" shouldn't it be "Molecular target for..." or "Target molecules for..."?
Same applies in: "7. Target molecular for NIR-PIT in gastrointestinal cancers"
(Response) We have corrected both of these to " Target molecules for...".
Similarly: Table 1. "weakly expression" shouldn't it be "weak expression" or "weakly expressed"?
(Response) We have corrected this to " weak expression".
I have no serious complaints about the presented publication and I recommend it for publication in Biomedicines.
(Response) Thank you very much for reviewing our article.
Reviewer 3 Report
The author reviews near-infrared photoimmunotherapy and summarizes it well overall.
The sentence in line 98 of the text, it may vary depending on the level of medical care in each country, but is it still true in the 2020s that 20-30% of lung cancer patients have airway symptoms? That said, compared to 2004 in the reference, medical equipment has changed and more often than not, the symptoms are detected before they appear. I think the references are outdated. Reference 49 in line 134 is a reference to apoptosis, but honestly it does not explain this sentence. Reference 52 is also outdated; lung cancer treatment has changed since 1991. In chapter 4, the author lists target molecules, some of which have few specific examples and I did not understand why they listed them. Also, overall, the author seems to cite too many of his own papers.
Author Response
The author reviews near-infrared photoimmunotherapy and summarizes it well overall.
The sentence in line 98 of the text, it may vary depending on the level of medical care in each country, but is it still true in the 2020s that 20-30% of lung cancer patients have airway symptoms? That said, compared to 2004 in the reference, medical equipment has changed and more often than not, the symptoms are detected before they appear. I think the references are outdated. Reference 49 in line 134 is a reference to apoptosis, but honestly it does not explain this sentence. Reference 52 is also outdated; lung cancer treatment has changed since 1991. In chapter 4, the author lists target molecules, some of which have few specific examples and I did not understand why they listed them. Also, overall, the author seems to cite too many of his own papers.
(Response)
Thank you for your review.
We have changed some references to newer ones.
The sentence in line 98 of the text, it may vary depending on the level of medical care in each country, but is it still true in the 2020s that 20-30% of lung cancer patients have airway symptoms? That said, compared to 2004 in the reference, medical equipment has changed and more often than not, the symptoms are detected before they appear
(Response)
Regarding the frequency of occurrence in the paper about CAO, the citation I used is used in a
recent paper, but it is indeed an older citation. we have changed it to the most recently surveyed
paper. Ref.29
Reference 49 in line 134 is a reference to apoptosis, but honestly it does not explain this sentence
(Response)
Replaced with citations that support showing that PDT and chemotherapy take the form of apoptosis-induced cell death and that it is delayed effect with no immediate effect.
Ref.49,50
Reference 52 is also outdated; lung cancer treatment has changed since 1991.
→
(Response)
The citation was old.
But regardless of therapeutic developments, malignant pleural effusions are still a symptom of advanced stages of lung cancer and their cause is still frequently attributed to pleural dissemination of lung cancer. We have replaced it with a recent citation for malignant pleural effusion. Ref.52
In chapter 4, the author lists target molecules, some of which have few specific examples and I did not understand why they listed them
→
(Response)
All target molecules expressed in lung cancer that have been reported to benefit from PIT in mouse models are listed. Some of them have not been studied in detail in lung cancer, but we hope that they will help further research in the future.
Also, overall, the author seems to cite too many of his own papers.
→
(Response)
Dr. Kobayashi is the inventor of NIR-PIT. He appears as an author on numerous papers because he collaborated or technically advised many NIR-PIT studies that were conducted all over the world. Therefore, this review focusing on NIR-PIT inevitably contains many citations.